# Sex-Specific Differences in Adipose IRF5 Expression and Its Association with Inflammation and Insulin Resistance in Obesity

**DOI:** 10.3390/ijms26178229

**Published:** 2025-08-25

**Authors:** Shihab Kochumon, Noelle Benobaid, Ashraf Al Madhoun, Shaima Albeloushi, Nourah Almansour, Fatema Al-Rashed, Sardar Sindhu, Fahd Al-Mulla, Rasheed Ahmad

**Affiliations:** 1Immunology and Microbiology Department, Dasman Diabetes Institute, Kuwait City 15462, Kuwait; shihab.kochumon@dasmaninstitute.org (S.K.); noelle.benobaid@dasmaninstitute.org (N.B.); shaima.albeloushi@dasmaninstitute.org (S.A.); nourah.almansour@dasmaninstitute.org (N.A.); fatema.alrashed@dasmaninstitute.org (F.A.-R.); sardar.sindhu@dasmaninstitute.org (S.S.); 2Animal and Imaging Core Facilities, Dasman Diabetes Institute, Dasman 15462, Kuwait; ashraf.madhoun@dasmaninstitute.org; 3Translational Research Department, Dasman Diabetes Institute, Kuwait City 15462, Kuwait; fahd.almulla@dasmaninstitute.org

**Keywords:** adipose tissue IRF5, FBG, HbA1c, HOMA-IR, inflammatory markers, obesity

## Abstract

Interferon regulatory factor 5 (IRF5) plays a pivotal role in innate immune responses and macrophage polarization. Although its role in obesity-associated inflammation has been described, sex-specific differences in adipose IRF5 expression and its association with immune and metabolic markers remain poorly defined. To evaluate sex-specific associations between adipose tissue (AT) IRF5 expression and key inflammatory and metabolic markers in overweight and obese individuals. Subcutaneous AT samples from overweight/obese male and female subjects were analyzed for IRF5 expression using quantitative reverse transcription-polymerase chain reaction (qRT-PCR). Correlation and multiple linear regression analyses were performed to identify its associations with inflammatory gene expression and metabolic parameters including insulin, glucose, HOMA-IR, and adipokines. RF5 gene and protein levels were significantly elevated in the AT of overweight/obese females compared to males (*p* < 0.0001), with expression increasing progressively with BMI in females but not in males. Despite these sex-dependent expression levels, IRF5 demonstrated consistent, sex-independent positive correlations with several core immune and inflammatory markers, including CCR5, CD11c, CD16, CD163, FOXP3, RUNX1, and MyD88. However, distinct sex-specific patterns emerged: in males, IRF5 correlated positively with classical pro-inflammatory markers such as IL-2, IL-6, IL-8, TNF-α, and IRAK1; whereas in females, IRF5 was associated with a broader array of immune markers, including chemokines (CCL7, CXCL11), pattern recognition receptors (TLR2, TLR8, TLR9), and macrophage markers (CD68, CD86), along with anti-inflammatory mediators such as IL-10 and IRF4. Notably, IRF5 expression in overweight/obese males, but not females, was significantly associated with metabolic dysfunction, showing positive correlations with fasting blood glucose, HbA1c, insulin, and homeostatic model assessment for insulin resistance (HOMA-IR) levels. Multiple regression analyses revealed sex-specific predictors of IRF5 expression, with metabolic (HOMA-IR) and inflammatory (IRAK1, MyD88) markers emerging in males, while immune-related genes (RUNX1, CD68, CCL7, MyD88) predominated in females. These findings underscore a sex-divergent role of IRF5 in AT, with implications for differential regulation of immune-metabolic pathways in obesity and its complications.

## 1. Introduction

Obesity is characterized by a chronic, low-grade inflammatory state that contributes to the development of insulin resistance and other metabolic comorbidities [1]. Adipose tissue (AT) inflammation, particularly the accumulation and activation of pro-inflammatory immune cells, is now recognized as a key mediator linking excess adiposity to metabolic dysfunction [2,3,4]. Importantly, clinical and epidemiological studies have consistently demonstrated sex-related differences in both the immune and metabolic consequences of obesity. While women generally have a higher percentage of body fat, men are more susceptible to visceral fat accumulation and its associated metabolic risks [5,6,7,8,9,10].

Sex hormones and sex-specific immune regulation may contribute to these differences, yet the molecular mediators driving sex-dependent AT inflammation remain insufficiently defined. One candidate of particular interest is interferon regulatory factor 5 (IRF5), a transcription factor that plays a pivotal role in innate immune signaling. IRF5 is best known for its role in type I interferon signaling and antiviral responses [11,12,13] However, emerging evidence positions IRF5 as a key modulator of macrophage polarization and inflammatory metabolism. IRF5 promotes M1 macrophage differentiation, while IRF4 supports the anti-inflammatory M2 state [14,15,16] In a landmark study, Dalmas et al. showed that mice lacking IRF5 in myeloid cells developed obesity when fed a high-fat diet but avoided metabolic complications. These IRF5-deficient mice displayed healthier adipose tissue expansion (subcutaneous rather than visceral), reduced inflammation, and enhanced insulin sensitivity. Notably, IRF5 directly suppressed TGFB1—a factor promoting tissue remodeling—in adipose tissue macrophages (ATMs), highlighting its role as a checkpoint in obesity-associated inflammation [17]. IRF5 promotes the polarization of macrophages toward a pro-inflammatory M1 phenotype and controls the expression of several cytokines involved in metabolic inflammation, including TNF-α, IL-6, and IL-12 [2,14,17,18]. Elevated IRF5 expression in AT has been associated with increased macrophage infiltration, enhanced Toll-like receptor (TLR) signaling, and impaired insulin sensitivity [14,17]. In humans, IRF5 expression in adipose tissue correlates robustly with metabolic inflammation and T2D [19]. In one study of diabetic obese patients, IRF5 gene and protein levels were elevated compared to lean individuals, and positively associated with a host of inflammatory markers, including TNF-α, IL-18, IL-23A, TLRs, MyD88, NF-κB, and macrophage markers CD11c and CD68 [20]. Another study confirmed increased IRF5 expression in obese versus lean subjects and found strong correlations with body mass index and body fat percentage [21]. These data substantiate IRF5 as a biomarker and potential driver of metabolic inflammation in obesity and T2D.

Despite these findings, few studies have addressed whether IRF5 expression and its downstream inflammatory network differ by sex in human obesity [22] Given the known sexual dimorphism in AT immune cell composition and metabolic response, understanding the sex-specific regulation and functional impact of IRF5 is of both mechanistic and clinical relevance. In this study, we aimed to investigate the sex-related variation in AT IRF5 expression and its relationship with inflammatory markers and insulin resistance in overweight and obese individuals. Using gene expression profiling of adipose biopsies and plasma metabolic measurements, we evaluated the correlation patterns between IRF5, immune-related genes, and metabolic indicators in men and women. These findings provide new insight into the immunometabolic role of IRF5 and its potential broad clinical implications, including the importance of understanding sex-specific differences (e.g., for personalized medicine or risk stratification) and its potential as a sex-specific biomarker or therapeutic target in obesity-related metabolic disease, such as diabetes and tissue inflammation.

## 2. Results

### 2.1. Sex-Related Differences in Adipose IRF5 Expression

In AT, the gene expression of IRF5 was significantly elevated in overweight and obese women compared to males (*p* < 0.0001), showing an 1.6-fold increase in females; suggesting a potential sex-linked regulatory mechanism that amplifies inflammatory gene expression in female adipose tissue (Figure 1A). Immunofluorescent analysis revealed a significant two-fold increase in IRF5 protein levels in AT of females compared to males (Figure 1B,C).

Next, we examined IRF5 transcript levels in relation to BMI by categorizing the samples into three groups: lean, overweight, and obese. In males, IRF5 expression in AT showed no significant differences among the BMI categories (*p* = 0.4027; Figure 1D). In contrast, overweight and obese females exhibited significantly higher IRF5 transcript levels compared to lean females (Figure 1E) These findings suggest that IRF5 gene expression is prominently associated with obesity in females.

### 2.2. Sex-Independent Correlation of IRF5 with Core Immune and Inflammatory Markers

To assess AT IRF5 crosstalk with inflammatory markers, correlation analyses were performed in overweight and obese -male and -female participants (*n* = 24 per gender). The findings revealed a combination of shared and sex-specific associations. However, several key correlations were consistent across genders, indicating a sex-independent pattern. Specifically, in both males and females, IRF5 expression demonstrated statistically significant positive correlations with a set of immune and inflammatory markers. These included the inflammatory chemokine receptor CCR5, as well as macrophage-associated markers CD11c, CD16, and CD163 (Table 1). Furthermore, in both genders, IRF5 expressions were positively correlated with the immune-regulatory transcription factors FOXP3 and RUNX1, as well as the TLR adaptor protein MyD88 (Table 1). The strength of these associations ranged from moderate (r = 0.434) to strong (r = 0.711) with all comparisons showing *p* < 0.05, indicating robust correlations. These consistent associations observed in both sexes suggest that IRF5 is part of a conserved immuno-inflammatory signature, regardless of biological sex. It supports the notion that IRF5 may play a central role in controlling immune and inflammatory responses in the context of obesity, independent of sex-related biological variation.

### 2.3. Sex-Specific Associations of AT IRF5 with Inflammatory Markers

Interestingly, the two genders exhibited a distinct pattern of correlations between IRF5 transcripts and various inflammatory markers in the AT (Table 1). In males, IRF5 gene expression levels displayed a significant positive association with the expression levels of the studied inflammatory factors (Table 1), including the interleukins IL-2, and IL-8, TNF-α, the dendritic cell marker CD141 and the interleukin kinase IRAK1 (r values ranged between 0.413 to 0.548; *p* < 0.05; Table 1); and to a less extent with that of IL6, IL23A, CCL5 and CCR2 (r ≥ 0.370; and *p* ranged between 0.068 to 0.077). On the other hand, IRF5’s correlation analysis in females a broader significant correlation with all studied inflammatory markers, reflecting a complex immunomodulatory profile (Table 1). In the latter group, positive associations were noticeable between IRF5 and the chemokines CCL7 (r = 0.530, *p* = 0.009) and CXCL11 (r = 0.431, *p* = 0.040), which are known to recruit monocytes and T-cells, respectively, to sites of inflammation. Moreover, IRF5 expression was positively associated with several pattern recognition receptors (PRRs), including dectin-1 (r = 0.441, *p* = 0.040), TLR2 (r = 0.549, *p* = 0.012), TLR8 (r = 0.751, *p* < 0.0001), and TLR9 (r = 0.432, *p* = 0.035, (Table 1). These receptors are key components of the innate immune system, responsible for detecting pathogen-associated molecular patterns and initiating immune responses. Additionally, IRF5 showed strong positive correlations with macrophage markers CD68 (r = 0.701, *p* < 0.0001) and CD86 (r = 0.637, *p* = 0.001), suggesting an association with activated macrophage populations within AT. Interestingly, IRF5 also correlated positively with the anti-inflammatory cytokine IL-10 (r = 0.542, *p* = 0.008) and the transcription factor IRF4 (r = 0.414, *p* = 0.044), which is involved in immune regulation and macrophage polarization (Table 1). In addition, partial correlation analysis of CXCL10, CCR1, CCR2, and TLR10 (r ≥ 0.369; and *p* ranged between 0.051 to 0.085, Table 1). These data suggest that IRF5 may contribute to both pro-inflammatory and regulatory pathways in female AT, indicating a more pronounced immunological role.

### 2.4. Increased AT IRF5 Gene Expression in Obesity Correlates with Diabetes Markers in Males

Since diabetes is a notable obesity complication, we were interested in studying the impact of the observed increase in IRF5 gene on diabetes status of overweight/obese individuals in our cohort. Notably, in overweigh/obese males, the AT IRF5 gene expression showed a significant positive correlation with the plasma levels of fasting blood glucose (FBG) (r = 0.546, *p* = 0.007; Figure 2A), and Hb1Ac (r = 0.510, *p* = 0.010; Figure 2B). In addition, males showed positive correlations between IRF5, and the plasma insulin levels (r = 0.617, *p* = 0.014), homeostatic model assessment for insulin resistance (HOMA-IR) levels, r = 0.657, *p* = 0.008) and as well as a negative correlation with adiponectin (r = −0.543), although the latter showed only moderate statistical significance (*p* = 0.068, Table 2). Alternatively, no significant correlations were observed between AT IRF5 transcripts and the diabetic markers in overweigh/obese females (*p* > 0.08, Figure 2C,D). The expression of IRF5 in females also showed a negative correlation with most of the studied biochemical parameters, particularly with plasma levels of insulin and HOMA-IR (Table 2). Together, these data suggest a potential role for IRF5 in glucose metabolism impairment and insulin resistance in men, whereas in women leaning more toward immune signaling than metabolic regulation.

### 2.5. Independent Predictors of IRF5 Expression

Parameters showing a significant association with the elevated IRF5 level were further evaluated in a multiple stepwise regression analysis (Table 1 and Table 2). Multiple linear regression analyses identified distinct predictors of IRF5 gene expression in each sex (Table 3). In males, IRAK1 (β = 0.570, *p* = 0.006), MyD88 (β = 0.512, *p* = 0.003) and HOMA-IR (β = 7.548, *p* = 0.001) were independent predictors of IRF5 expression. In contrast, IRF5 gene expression in females was independently associated by RUNX1 (β = 0.399, *p* = 0.005), CD68 (β = 0.333, *p* = 0.016), CCL7 (β = 0.313, *p* = 0.010), and MyD88 (β = 0.286, *p* = 0.030).

## 3. Discussion

This study reveals a differential association of IRF5 with metabolic dysfunction in males versus females despite higher expression in females, particularly in the contexts of obesity. We demonstrate that IRF5 expression is significantly elevated in the AT of overweight and obese women compared to men. Despite this elevation, IRF5 shows stronger associations with insulin resistance and metabolic dysfunction in men, suggesting sex-specific functional consequences of IRF5 activity in obesity.

Our findings confirm and expand upon earlier reports implicating IRF5 as a central regulator of AT inflammation and macrophage polarization [6,17]. Notably, we observed a strong correlation between IRF5 and canonical inflammatory markers such as CCR5, CD11c, CD16, CD163, and FOXP3 in both sexes. These shared associations suggest that IRF5 may serve as a core inflammatory node irrespective of sex, aligning with its known role in promoting pro-inflammatory M1 macrophage phenotypes [14]. However, the divergence in sex-specific correlations highlights potential biological differences in IRF5-mediated immune signaling. In males, IRF5 was positively associated with classical pro-inflammatory cytokines (IL-2, IL-8, TNF-α), as well as metabolic markers including insulin, HbA1c, and HOMA-IR. These associations support previous studies linking IRF5 to metabolic dysfunction and insulin resistance [14,17], and further suggest that in men, IRF5 may play a more direct role in linking immune activation to metabolic impairment. Conversely, in females, IRF5 showed significant correlations with IL-10 and IRF4, both of which are recognized as anti-inflammatory mediators [15,23]. This may reflect a more pronounced or compensatory immune environment in women, where elevated IRF5 expression is counterbalanced by concurrent anti-inflammatory signaling [24]. Furthermore, the association of IRF5 with the Toll-like receptors (TLR2, TLR8, and TLR9), as well as with macrophage markers CD68, CD86, and the chemokine CCL7, suggests an enhanced pattern recognition and antigen presentation profile in females. This immune activation may contribute to local inflammation without directly impairing insulin sensitivity, consistent with previous evidence indicating sex-specific immune regulation in metabolic tissues [25,26].

The sex-specific predictors identified through regression analyses further highlight distinct regulatory networks governing IRF5 expression in AT. In men, predictors such as IRAK1 and MyD88, key signaling molecules downstream of TLRs, were strongly associated with IRF5 expression, reinforcing a pro-inflammatory signaling axis consistent with TLR-IRF5 pathway activation [27,28]. Additionally, HOMA-IR emerged as independent predictors, suggesting that IRF5 is closely integrated into the metabolic-inflammatory circuit in male AT, as previously observed in obesity-related insulin resistance [17,29]. In contrast, in women, primary predictors of IRF5 expression included RUNX1, CD68, CCL7, and MyD88, indicating a transcriptional and chemokine-driven mode of regulation. These associations may reflect enhanced immune cell recruitment, macrophage differentiation, and tissue remodeling, consistent with RUNX1’s role in myeloid lineage development and immune cell activation [30]. Collectively, these findings point to sex-specific immune regulatory mechanisms shaping the function and expression of IRF5 in AT.

Interestingly, although IRF5 expression was significantly higher in women, this did not translate into measurable impairment of insulin sensitivity or glycemic control. This observation may help explain the paradox of heightened inflammation in women with obesity, yet a relative preservation of insulin sensitivity, a phenomenon documented in both clinical and epidemiological studies [5,31,32]. Such sex-specific metabolic resilience may stem from several interconnected factors. Women typically have a different distribution of adipose tissue, a distinct immune cell profile within that tissue, and hormonal influences—particularly from estrogen—that together can shape the inflammatory environment. These factors may allow inflammation to occur in adipose tissue without causing the same degree of disruption to insulin signaling seen in males [33,34].

Taken together, our data underscore the importance of considering sex as a biological variable in immunometabolic research. IRF5 appears to be a sex-dependent modulator of AT inflammation with differential implications for metabolic health. These findings suggest that therapeutic targeting of IRF5 or its upstream regulators may require sex-specific strategies to achieve optimal outcomes in metabolic disease The sex-specific associations we report have important therapeutic implications. In males, where IRF5 strongly correlates with insulin resistance and metabolic dysfunction, therapeutic approaches that target IRF5 directly or modulate its upstream signaling pathways (e.g., TLR–MyD88–IRAK1) may help mitigate obesity-associated metabolic impairment. In contrast, in females, elevated IRF5 expression is accompanied by compensatory anti-inflammatory mediators, suggesting that therapeutic strategies should aim to preserve this balance rather than broadly suppress IRF5 activity. This sex-dependent divergence underscores the need for personalized approaches in obesity and metabolic disease management, where pharmacological interventions, lifestyle modifications, and biomarker-driven risk stratification may need to be tailored according to sex-specific immunometabolic profiles.

## 4. Materials and Methods

### 4.1. Study Population

A total of 48 non-diabetic adult participants—comprising 24 males and 24 females—were recruited from the Dasman Diabetes Institute in Kuwait. Individuals were excluded if they had a history of serious medical conditions such as cardiovascular, pulmonary, renal, hepatic, or hematological diseases, as well as autoimmune disorders, malignancies, pregnancy, type 1 diabetes (T1D) or type 2 diabetes (T2D), to reduce potential confounding variables. Participants were stratified into three categories based on body mass index (BMI): lean (BMI < 25 kg/m^2^), overweight (BMI 25–30 kg/m^2^), and obese (BMI > 30 kg/m^2^). The clinical and demographic characteristics of the participants are summarized in Appendix A. All individuals provided written informed consent before participating in the study. Ethical approval was obtained from the Dasman Diabetes Institute’s Ethics Committee (Reference: RA 2010-003), which adheres to internationally accepted principles for research involving human subjects, in accordance with the Declaration of Helsinki (October 2013 revision).

### 4.2. Anthropometric and Physio-Clinical Measurements

Anthropometric and physio-clinical measurements were made including body weight, height, waist circumference, and systolic and diastolic blood pressure. Body weight was measured with portable electronic weighing scale and height was measured by using height measuring bars. Waist circumference was measured by using constant tension tape at the end of a normal expiration with arms relaxed at sides. Body composition analyzer (IOI 353, Spectronix Medical Trading, Seoul, South Korea) was used to measure whole body composition including the body fat percentage, soft lean mass, and total body water. The following formula was used to calculate the BMI: BMI = Body weight (Kg)/Height (m^2^). Peripheral blood/fat biopsy samples were collected from the study participants after overnight (10 h minimum) fasting and samples were analyzed to determine the fasting plasma glucose, glycated hemoglobin (HbA1c), fasting serum insulin, and serum lipids levels. Glucose and lipids were detected by using Siemens dimension RXL chemistry analyzer (Diamond Diagnostics, Holliston, MA, USA). HbA1c was detected with Variant device (BioRad, Hercules, CA, USA). To obtain plasma, anticoagulated blood was centrifuged at 1200× *g* for 10 min and collected plasma was aliquoted and stored at −80 °C until use. Plasma triglycerides were determined using commercial kit (Intra-assay CV% = 0.93; Inter-assay CV% = 3.05) (Chema Diagnostica, Monsano, Italy). In our study, plasma adiponectin levels were measured using the Human Adiponectin/Acrp30 Magnetic Luminex^®^ Performance Assay (Catalog #: LOBM1065, R&D Systems, Austin, TX, USA) This method was selected due to its high sensitivity (6.4 pg/mL). All assays were performed following the manufacturers’ instructions.

### 4.3. Adipose Tissue Collection and Processing

Abdominal subcutaneous AT samples (~0.5 g) were obtained using a standard biopsy technique, performed lateral to the umbilicus, as previously described [35]. In brief, the periumbilical region was cleansed with alcohol and locally anesthetized using 2 mL of 2% lidocaine. A small skin incision (~0.5 cm) was then made to access and excise the subcutaneous AT. The collected AT was sectioned into smaller fragments, rinsed thoroughly with cold phosphate-buffered saline (PBS), and then fixed in 4% paraformaldehyde for 24 h. Following fixation, tissues were embedded in paraffin for histological analysis. In parallel, a portion of freshly harvested AT (approximately 50–100 mg) was placed in RNAlater™ solution to preserve RNA integrity and stored at −80 °C for future molecular analysis.

### 4.4. Real-Time Reverse-Transcription Polymerase Chain Reaction (RT-PCR)

To determine gene expression, total RNA was purified from the AT following the manufacturer’s instructions (RNeasy kit, Qiagen, Valencia, CA, USA) as described [36]. The isolated RNA was quantified using Epoch Spectrophotometer (BioTek, Winooski, VT, USA) and RNA quality was evaluated by formaldehyde-agarose gel electrophoresis. One microgram of each RNA sample was reverse transcribed into cDNA using random hexamer primers and TaqMan RT reagents (High Capacity, cDNA RT Kit; Applied Biosystems, Foster City, CA, USA). Then, cDNA (50 ng) was amplified by using TaqMan Gene Expression MasterMix (Applied Biosystems) together with target gene-specific 20× TaqMan Gene Expression Assays (Applied Biosystems) containing forward/reverse primers (Appendix A) and target-specific TaqMan minor groove binder (MGB) probe labeled with 6-fluorescein amidite (FAM) dye at 5′ and with non-fluorescent quencher (NFQ)-MGB at 3′ end of the probe for 40 cycles of PCR amplification using 7500 Fast Real-Time PCR System (Applied Biosystems, CA, USA). Each thermal cycle included heating at 95 °C for 15 s for denaturation, then heating at 60 °C for 1 min for annealing/extension, followed by heating at 50 °C for 2 min for uracil DNA glycosylase activation and later, heating at 95 °C for 10 min for AmpliTaq Gold enzyme activation. The expression of glyceraldehyde 3-phosphate dehydrogenase (GAPDH) was used as internal control to normalize differences in individual samples compared to control sample (lean AT). Target gene expression (relative mRNA expression) was calculated using the 2^−ΔΔCt^ method and was expressed as fold change (mean ± SEM) over the average GAPDH expression taken as one.

### 4.5. Immunofluorescence Microscopy

Formalin-fixed AT were paraffin-embedded and micro-sectioned by microtomy (8 μm thick) and labelled with immunofluorescent antibodies, as previously described [37,38]. Briefly, on day-1, slides were deparaffinized and rehydrated; then antigen retrieval was carried out. Slides were blocked with 10% goat serum and then incubated overnight at 4 °C with rabbit anti-human IRF5 antibody (1:250 dilution, ab181553, Abcam, Cambridge, UK). On day-2, slides were incubated for one-hour with goat anti-rabbit Alexa flour-488 (1:200, A11034, ThermoFisher, Waltham, MA, USA); then slides were incubated with rabbit anti-human Perilipin antibody (1:150 dilution, 9349, Cell Signaling) overnight at 4 °C. On day-3, slides were incubated with goat anti-rabbit Alexa Fluor-546 (1:200, A10040, ThermoFisher) for one hour and mounted with ProLong Gold antifade mounting medium with DAPI (P36935, Invitrogen, Waltham, MA, USA). Immunostained sections were visualized using an Olympus BX53 upright microscope connected to an Olympus DP73 camera (Olympus, Tokyo, Japan) and an inverted Zeiss LSM710 spectral confocal microscope (Carl Zeiss, Gottingen, Germany). IRF5 positive areas were digitally quantified using ImageJ software (Fiji is just ImageJ version 2.9.0/1.53t, College Park, MD, USA), which measured the staining signal intensity as an output of pixel grey value (px GV), ranging from 0 to 255 [39,40].

### 4.6. Statistical Analysis

Statistical analysis was performed using GraphPad Prism software, version 10.5.0 (La Jolla, San Diego, CA, USA) and SPSS for Windows version 19.01 (IBM SPSS Inc., Armonk, NY, USA). Data are shown as mean ± standard deviation values, unless otherwise indicated. Unpaired Student *t*-test was used to compare means between groups. Correlation and multiple regression analysis were performed to determine association between different variables. For all analyses, *p*-value < 0.05 was considered significant.

## 5. Conclusions

This study demonstrates that IRF5 is differentially expressed and functionally linked to distinct inflammatory and metabolic pathways in a sex-specific manner. In men, IRF5 may serve as a biomarker and potential association with insulin resistance, while in women, it may reflect broader immune activation with less direct metabolic consequence. These insights may inform future therapeutic approaches for obesity associated with metabolic disorders.

## 6. Limitations and Future Directions

The sample size, while well-balanced between sexes, is relatively modest and limited to non-diabetic individuals, which may restrict the generalizability of these findings. Other limitations include the absence of females’ menopausal status, and the hormonal variation between genders. Future studies in larger, more diverse cohorts, including individuals with T2D, are warranted. Moreover, functional validation of IRF5’s role in macrophage subtypes and insulin signaling pathways would further clarify its mechanistic contributions. Moreover, a valuable approach is to investigate the hormonal and genetic mechanisms driving the observed sex differences, including the influence of sex hormones, X-linked genes, and gene–environment interactions on IRF5 activity and downstream immunometabolic pathways.

## Figures and Tables

**Figure 1 ijms-26-08229-f001:**
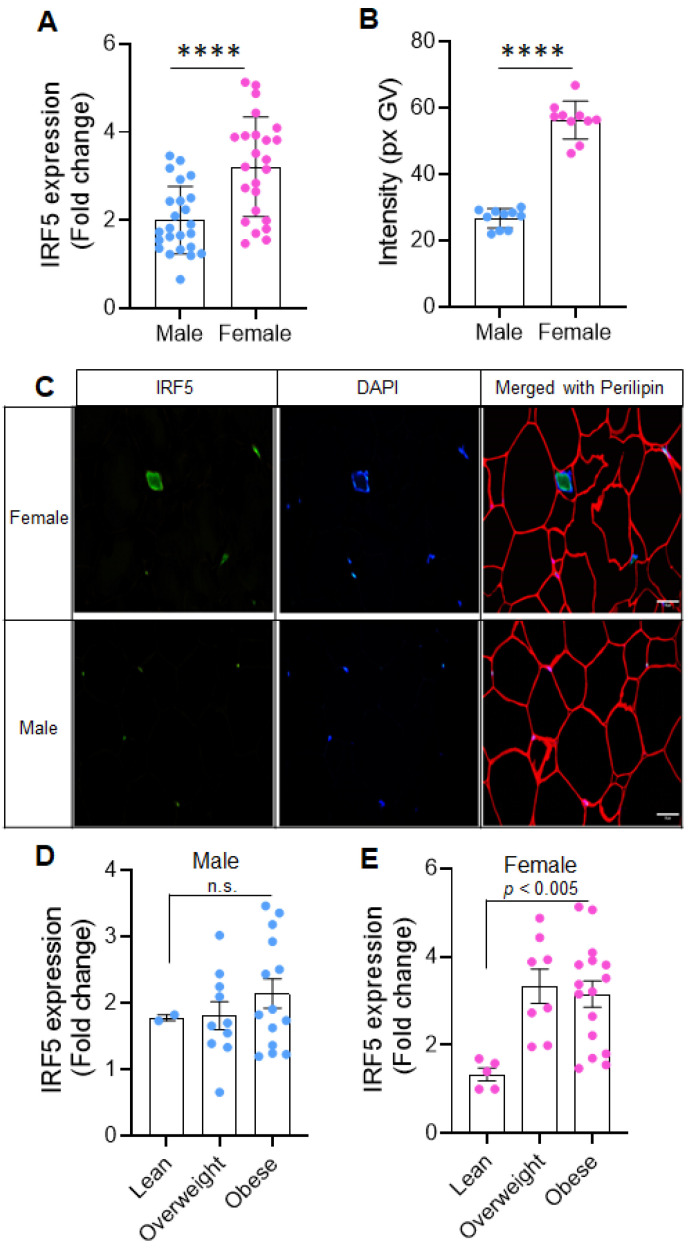
Sex-related differences in adipose tissue IRF5 expression and its association with obesity. (**A**) Quantitative analysis of IRF5 mRNA expression in adipose tissue from males and females reveals significantly higher expression in females compared to males (*p* < 0.0001), indicating sex-dependent differences in IRF5 transcript levels. (**B**) Protein expression analysis of IRF5 by immunofluorescence confirms increased levels in adipose tissue from females versus males, as shown by box-and-whisker plot quantification of fluorescent intensity. (**C**) Representative immunofluorescence images of adipose tissue sections stained for IRF5 (green), nuclei (DAPI, blue), and cell membranes (Phalloidin, red) in male (bottom row) and female (top row) samples. Increased nuclear and cytoplasmic IRF5 expression is evident in female adipocytes. Scale bar = 20 μm. (**D**,**E**) Analysis of IRF5 transcript levels across BMI categories. In males (**D**), IRF5 expression does not significantly differ among lean, overweight, and obese individuals. In females (**E**), IRF5 expression increases progressively with BMI, with significantly higher levels in overweight and obese women compared to lean women (*p* < 0.05), indicating a strong association between IRF5 and obesity in females. **** *p* ≤ 0.0001.

**Figure 2 ijms-26-08229-f002:**
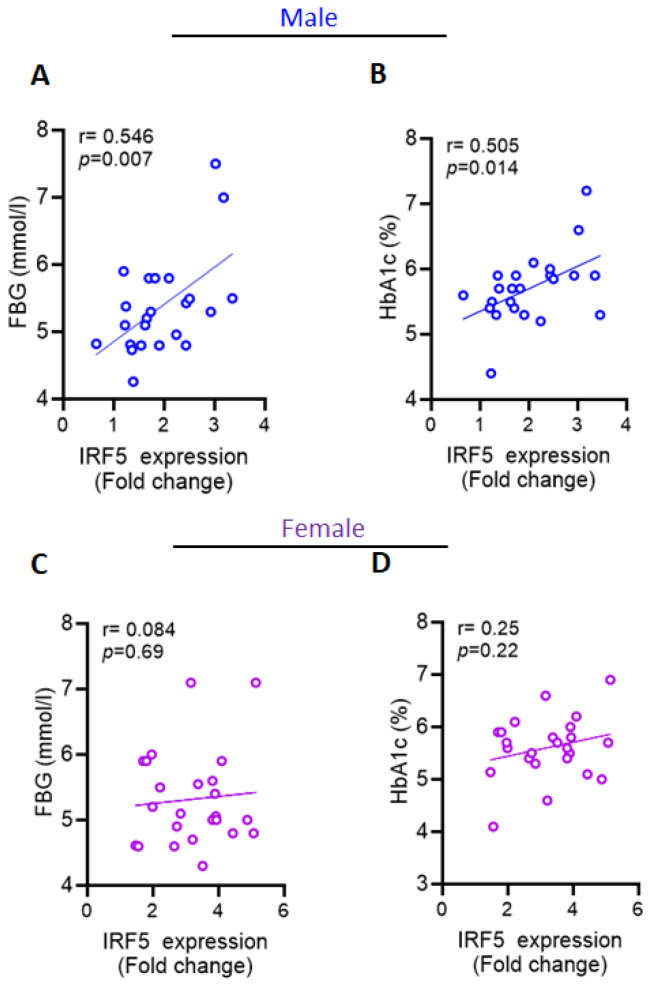
Sex-specific associations between adipose tissue IRF5 expression and metabolic markers. (**A**,**B**) In overweight/obese male participants, adipose tissue (AT) IRF5 transcript levels showed a significant positive correlation with (**A**) fasting blood glucose (FBG) (r = 0.546, *p* = 0.007) and (**B**) HbA1c levels (r = 0.510, *p* = 0.010). (**C**,**D**) In contrast, no significant correlations were observed between AT IRF5 transcript levels and (**C**) fasting blood glucose or (**D**) HbA1c in overweight/obese female participants. Each data point represents an individual subject. Blue indicates males and purple indicates females. Solid lines represent linear regression fit.

**Table 1 ijms-26-08229-t001:** Correlation between IRF5 expression in adipose tissue and inflammatory markers in male and female subjects or IRF5 expression–inflammation link in adipose tissue: male vs. female Pearson correlation, * *p* < 0.05, ** *p* < 0.01.

Inflammatory Markers	Male (Overweight + Obese)		Female (Overweight + Obese)	
	r-Value	*p*-Value	r-Value	*p*-Value
Cytokines				
IL2	0.413	0.045 *	0.277	0.212
IL6	0.370	0.075	0.068	0.765
IL8	0.458	0.032 *	0.208	0.352
IL10	0.107	0.635	0.542	0.008 **
IL33	0.148	0.489	0.114	0.613
TNFα	0.474	0.019 *	0.359	0.101
TGFβ	0.207	0.343	0.079	0.727
CC and CXC chemokine ligands and receptors				
CCL2	0.251	0.236	0.197	0.355
CCL5	0.394	0.077	0.265	0.273
CCL7	0.255	0.241	0.530	0.009 **
CCL19	0.368	0.084	0.195	0.396
CXCL10	−0.007	0.975	0.410	0.052
CXCL11	0.137	0.534	0.431	0.040 *
Dectin-1	0.328	0.126	0.441	0.040 *
CCR1	0.031	0.895	0.369	0.083
CCR2	0.403	0.070	0.375	0.085
CCR5	0.434	0.034 *	0.440	0.032 *
Macrophage markers				
CD11c	0.677	0.000 **	0.639	0.001 **
CD16	0.578	0.003 **	0.44	0.040 *
CD68	0.394	0.057	0.701	0.000 **
CD86	0.195	0.385	0.637	0.001 **
CD141	0.464	0.026 *	0.032	0.883
CD163	0.711	0.000 **	0.534 **	0.009 **
Toll-like receptors (TLRs) signaling cascade				
TLR2	0.108	0.633	0.549	0.012 *
TLR4	0.385	0.094	0.257	0.262
TLR8	0.169	0.431	0.751	0.000 **
TLR9	0.355	0.089	0.432	0.035 *
TLR10	0.019	0.931	0.442	0.051
MyD88	0.495	0.014 *	0.584 **	0.003
IRAK1	0.548	0.006 **	0.304	0.158
TRAF6	0.087	0.688	0.087	0.687
Transcription factors				
FOXP3	0.548	0.006 **	0.464	0.030 *
RUNX1	0.507	0.011 *	0.681	0.000 **
IRF4	0.110	0.626	0.414	0.044 *
IRF3	0.383	0.096	0.265	0.246

**Table 2 ijms-26-08229-t002:** Relationship Between Adipose Tissue IRF5 Expression and Metabolic Markers by Sex. Pearson correlation, * *p* < 0.05, ** *p* < 0.01.

Metabolic Markers	Male (Overweight + Obese)		Female (Overweight + Obese)	
	r-Value	*p*-Value	r-Value	*p*-Value
Weight (kg)	0.366	0.079	−0.119	0.579
Height (cm)	0.347	0.097	−0.158	0.461
BMI (kg/m^2^)	0.207	0.333	−0.012	0.955
PBF (number)	0.179	0.463	0.002	0.992
Waist (cm)	0.412	0.071	−0.075	0.741
Cholestrol (mmol/L)	0.121	0.573	0.136	0.525
HDL (mmol/L)	−0.171	0.424	−0.203	0.341
LDL (mmol/L)	0.014	0.947	0.230	0.279
TGL (mmol/L)	0.304	0.148	−0.257	0.226
FBG (mmol/L)	0.546	0.007 **	0.084	0.696
HbA1c (%)	0.505	0.014 *	0.255	0.229
Insulin (mU/L)	0.617	0.014 *	−0.363	0.167
HOMA-IR	0.657	0.008 **	−0.283	0.289
Adiponectin (ng/mL)	−0.543	0.068	−0.371	0.130
RANTES (ng/mL)	0.515	0.086	0.073	0.774

BMI, body mass index; PBF, Peripheral Blood Film; HDL, high-density lipoprotein; LDL, low-density lipoprotein; TGL, Triglycerides; FBG, fasting blood glucose; HbA1c, glycated hemoglobin C; HOMA-IR, homeostatic model assessment for insulin resistance.

**Table 3 ijms-26-08229-t003:** Multi Linear Regression analysis, with IRF5 as a dependent variable for each gender.

ANOVA (Sig) R^2^ = 082; *p* < 0.0001				
Predictor Variable	Male		Female	
	Scandalized Confinement (β)	*p*-Value	Scandalized Confinement (β)	*p*-Value
IRAK1	0.570	0.006	_	_
HOMA-IR	7.548	0.001	_	_
MyD88	0.512	0.003	0.288	0.030
RUNX1	_	_	0.399	0.005
CD68	_	_	0.333	0.016
CCL7	_	_	0.313	0.017

## Data Availability

The data presented in this study are available on request from the corresponding author.

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
