# Peer review of "Sex-Specific Differences in Adipose IRF5 Expression and Its Association with Inflammation and Insulin Resistance in Obesity"

_ijms, 2025, doi:10.3390/ijms26178229_

Round 1

Reviewer 1 Report

Comments and Suggestions for Authors

The authors show the sex-dependent role of IRF5 in adipose tissue and its connection with inflammatory and metabolic marker. The study design is simple and clear, and the data are presented in a well-structured manner. But the manuscript needs some revision before it can be published.

1) Kindly enhance clarity of figures (e.g., increase font sizes, use clearer legends). Figure 1A: Clarify whether individual dots represent biological replicates.

 2) Sex differences in adipose tissue inflammation may be influenced by menopausal status and hormonal variation, but this is not mentioned. Provide data on menopausal status or acknowledge it as a limitation.

3) The manuscript is generally well-written, but the Results and Discussion sections are overly descriptive and lack critical interpretation. Figures are cluttered and difficult to interpret quickly; some legends are too brief to stand alone.

 4) Some abbreviations (e.g., HOMA-IR, T2D) are used before being defined. Improve flow by consolidating redundant observations in the results section.

5) Discuss broader implications of sex-specific findings for therapeutic strategies.

6) Apply appropriate correction for multiple comparisons to Table 1 correlations. Report adjusted R² for regression models. Correct terminology in Table 3.

7) Define all abbreviations upon first use in the abstract and main text.

8) A few grammatical inconsistencies are present (e.g., singular/plural mismatch).

Comments on the Quality of English Language

The authors show the sex-dependent role of IRF5 in adipose tissue and its connection with inflammatory and metabolic marker. The study design is simple and clear, and the data are presented in a well-structured manner. But the manuscript needs some revision before it can be published.

1) Kindly enhance clarity of figures (e.g., increase font sizes, use clearer legends). Figure 1A: Clarify whether individual dots represent biological replicates.

 2) Sex differences in adipose tissue inflammation may be influenced by menopausal status and hormonal variation, but this is not mentioned. Provide data on menopausal status or acknowledge it as a limitation.

3) The manuscript is generally well-written, but the Results and Discussion sections are overly descriptive and lack critical interpretation. Figures are cluttered and difficult to interpret quickly; some legends are too brief to stand alone.

 4) Some abbreviations (e.g., HOMA-IR, T2D) are used before being defined. Improve flow by consolidating redundant observations in the results section.

5) Discuss broader implications of sex-specific findings for therapeutic strategies.

6) Apply appropriate correction for multiple comparisons to Table 1 correlations. Report adjusted R² for regression models. Correct terminology in Table 3.

7) Define all abbreviations upon first use in the abstract and main text.

8) A few grammatical inconsistencies are present (e.g., singular/plural mismatch).

Author Response

Comments and Suggestions for Authors

The authors show the sex-dependent role of IRF5 in adipose tissue and its connection with inflammatory and metabolic marker. The study design is simple and clear, and the data are presented in a well-structured manner. But the manuscript needs some revision before it can be published.

We thank the reviewer for the positive feedback on our study design and data presentation.

  • Kindly enhance clarity of figures (e.g., increase font sizes, use clearer legends). Figure 1A: Clarify whether individual dots represent biological replicates.

Authors’ response: We would like to thank the reviewer for the notification. We fixed Figure 1, including font and design.

  • Sex differences in adipose tissue inflammation may be influenced by menopausal status and hormonal variation, but this is not mentioned. Provide data on menopausal status or acknowledge it as a limitation.

Authors’ response: We would like to thank the reviewer for the notification. We do not menopausal status and hormonal variation data. The fact is acknowledged in the limitation sections. Please see lines 378-379, red font.

  • The manuscript is generally well-written, but the Results and Discussion sections are overly descriptive and lack critical interpretation. Figures are cluttered and difficult to interpret quickly; some legends are too brief to stand alone.

Authors’ response: We thank the reviewer for the helpful feedback. We have revised the Results and Discussion to include more critical interpretation, highlighting mechanistic implications and integrating findings with existing literature. Figures have been simplified  and legends expanded

  • Some abbreviations (e.g., HOMA-IR, T2D) are used before being defined. Improve flow by consolidating redundant observations in the results section.

Authors’ response: We thank the reviewer for pointing this out. We have revised the manuscript to ensure that all abbreviations (e.g., HOMA-IR, T2D) are defined at first mention for clarity. Additionally, we have improved the flow of the Results section by consolidating redundant observations and streamlining the text to avoid repetition, while maintaining accuracy and completeness of the findings.

  • Discuss broader implications of sex-specific findings for therapeutic strategies.

Authors’ response: We appreciate the reviewer’s suggestion. In line with this, we have expanded the final paragraph of the Discussion to explicitly address the broader therapeutic implications of our sex-specific findings.

  • Apply appropriate corrections for multiple comparisons to Table 1 correlations. Report adjusted R² for regression models. Correct terminology in Table 3.

Authors response: Corrected as suggested.

  • Define all abbreviations upon first use in the abstract and main text.

Authors response: Done as suggested.

  • A few grammatical inconsistencies are present (e.g., singular/plural mismatch).

Authors response: Corrected as suggested.

Reviewer 2 Report

Comments and Suggestions for Authors

This paper provides a significant contribution to the understanding of sex-specific differences in the IRF5 expression in adipose and its link to inflammation and insulin resistance in obesity. Although the study is well-designed and the findings are highly important to the field of immunometabolism, the authors need to address the following major comments before the article is accepted. Overall, the study is well-designed, and the data are consistent and satisfactory; however, several minor concerns need to be addressed before the paper is ready for publication.

Major comments

Introduction

Line 70-72: While the introduction effectively highlights the research gap, a brief mention of the potential broader clinical implications of understanding these sex-specific differences (e.g., for personalized medicine or risk stratification) could further enhance its impact. This could be added after the statement about the study's aim and its potential as a sex-specific biomarker or therapeutic target

Material and Methods: In the Study Population section, the authors should specify the number of male and female participants receiving any form of therapy for hypertension or lipid-lowering treatment. If applicable, the specific names of the medications used should also be included for clarity

Lines 107–108: The authors did not specify the name of the kit used for the determination of plasma adiponectin, nor did they provide any information regarding the specificity and sensitivity of the method.

Results

Figure 1A and 1B: For improved readability, especially in Figure 1A and 1B, considering slightly larger font sizes for y-axis labels ('IRF5 expression (Fold change)' and 'Intensity (px GV)') would be beneficial.

Table 1: In Table 1, maintaining consistent formatting for p-values (e.g., always showing three decimal places, or using '<0.001' for very small values) across all entries would enhance uniformity and precision.

Line 211-212: The statement "The strength of these associations ranged from moderate to strong (0.434 ≥ r ≤ 0.711, p < 0.05 for all comparisons)" could be slightly rephrased for clarity, perhaps to "The strength of these associations ranged from moderate (r = 0.434) to strong (r = 0.711), with all comparisons showing p < 0.05".

In lines 257-258, clarify the 'weak negative trend with adiponectin' in males. While the p-value (0.068) is provided, explicitly stating the r-value (-0.543) in the text would give the reader a better sense of the trend's strength, even if it's not statistically significant.

General comment,  please ensure that all figures and tables are referenced in the text in sequential order. A quick check confirms this is largely done, but a final pass would be beneficial.

Discussion:

Line 301-306: To further enhance the impact of the discussion, a brief introductory paragraph summarizing the most striking novel findings (e.g., the differential association of IRF5 with metabolic dysfunction in males vs. females despite higher expression in females) before delving into detailed interpretations could be considered. This could be placed after the initial summary of the study's findings.

Line 351-352: Please expand slightly on the implications of these sex-specific findings for personalized medicine approaches in obesity management, beyond just the direct targeting of IRF5, could add further value. This could be a sentence or two following the statement about sex-specific strategies for therapeutic targeting of IRF5.

Line 361-362: For the 'Limitations and Future Directions' section, consider adding a sentence about the specific future research avenues that directly build on the novel sex-specific findings, such as investigating the underlying hormonal or genetic mechanisms driving these differences.

Author Response

Reviewer 2

Comments and Suggestions for Authors

This paper provides a significant contribution to the understanding of sex-specific differences in the IRF5 expression in adipose and its link to inflammation and insulin resistance in obesity. Although the study is well-designed and the findings are highly important to the field of immunometabolism, the authors need to address the following major comments before the article is accepted. Overall, the study is well-designed, and the data are consistent and satisfactory; however, several minor concerns need to be addressed before the paper is ready for publication.

We would like to thank the reviewer for the compliments and insights. We appreciate the positive feedback.

Major comments

Introduction

Line 70-72: While the introduction effectively highlights the research gap, a brief mention of the potential broader clinical implications of understanding these sex-specific differences (e.g., for personalized medicine or risk stratification) could further enhance its impact. This could be added after the statement about the study's aim and its potential as a sex-specific biomarker or therapeutic target

We thank the Reviewer for complements. We added the suggested sentence into this paragraph. Please see Lines 72-75.

Material and Methods: In the Study Population section, the authors should specify the number of male and female participants receiving any form of therapy for hypertension or lipid-lowering treatment. If applicable, the specific names of the medications used should also be included for clarity

We have now clarified the use of medications within our study population. Specifically, five participants (three males and two females) were taking medications for hypertension at the time of data collection: Concor, Lipitor, and Aldomet. Among these, only two participants were using Lipitor, a lipid-lowering agent.

Lines 107–108: The authors did not specify the name of the kit used for the determination of plasma adiponectin, nor did they provide any information regarding the specificity and sensitivity of the method.

Thank you for the comment. In our study, plasma adiponectin levels were measured using the Human Adiponectin/Acrp30 Magnetic Luminex® Performance Assay (Catalog #: LOBM1065, R&D Systems, Austin, TX, USA )

This method was selected due to its high sensitivity (6.4 pg/mL). Please see lines 340-344.

Results

Figure 1A and 1B: For improved readability, especially in Figure 1A and 1B, considering slightly larger font sizes for y-axis labels ('IRF5 expression (Fold change)' and 'Intensity (px GV)') would be beneficial.

We would like to thank the reviewer for the notification. We fixed Figure 1, including font and design.

Table 1: In Table 1, maintaining consistent formatting for p-values (e.g., always showing three decimal places, or using '<0.001' for very small values) across all entries would enhance uniformity and precision.

We would like to thank the reviewer for the notification. We fixed Table 1, 2 and 3, accordingly. 

Line 211-212: The statement "The strength of these associations ranged from moderate to strong (0.434 ≥ r ≤ 0.711, p < 0.05 for all comparisons)" could be slightly rephrased for clarity, perhaps to "The strength of these associations ranged from moderate (r = 0.434) to strong (r = 0.711), with all comparisons showing p < 0.05".

We would like to thank the reviewer for the notification. The text was corrected accordingly (Lines 82-83, red fond). 

In lines 257-258, clarify the 'weak negative trend with adiponectin' in males. While the p-value (0.068) is provided, explicitly stating the r-value (-0.543) in the text would give the reader a better sense of the trend's strength, even if it's not statistically significant.

We would like to thank the reviewer for the notification. We corrected the sentence, (lines 190-193, red font). 

General comment, please ensure that all figures and tables are referenced in the text in sequential order. A quick check confirms this is largely done, but a final pass would be beneficial.

Figures and tables referencing in the text were checked.

Discussion:

Line 301-306: To further enhance the impact of the discussion, a brief introductory paragraph summarizing the most striking novel findings (e.g., the differential association of IRF5 with metabolic dysfunction in males vs. females despite higher expression in females) before delving into detailed interpretations could be considered. This could be placed after the initial summary of the study's findings.

We thank the reviewer for this notable statement, which we added to the paragraph, please see lines 305-306. 

Line 351-352: Please expand slightly on the implications of these sex-specific findings for personalized medicine approaches in obesity management, beyond just the direct targeting of IRF5, could add further value. This could be a sentence or two following the statement about sex-specific strategies for therapeutic targeting of IRF5.

We thank the reviewer for this notable statement. We improved the paragraph, please see lines 357-361, red font.

Line 361-362: For the 'Limitations and Future Directions' section, consider adding a sentence about the specific future research avenues that directly build on the novel sex-specific findings, such as investigating the underlying hormonal or genetic mechanisms driving these differences.

We thank the reviewer for this notable statement. We improved the paragraph, please see lines 375-378, red font.

Reviewer 3 Report

Comments and Suggestions for Authors

This manuscript by Kochumon et al. investigates the sex-specific expression of IRF5 in human adipose tissue and its associations with inflammation and insulin resistance in obese individuals. The authors analyze gene expression data from adipose biopsies, highlighting differential associations between IRF5 and metabolic parameters in females versus males. While the manuscript provides interesting observations, several key concerns related to interpretation, analytical depth, and translational context need to be addressed before the study can be considered for publication.

Major Comments

  1. The introduction lacks a detailed rationale for focusing specifically on IRF5 among the various immune regulators involved in adipose inflammation.
  2. The background does not clearly explain the biological plausibility for sex differences in IRF5 expression or signaling.
  3. Expand the introduction to justify IRF5 selection and elaborate on possible biological pathways and their involvements in inflammations.
  4. Please provide the Information about the sample collection (e.g., fasting status, comorbidities, or medications) which could confound IRF5 expression. These can strongly affect IRF5 expression and insulin sensitivity.
  5. Line 72: Clarify the meaning of “metabolic disease” obesity.
  6. In lines 173-174, authors should consider adding the specific fold change or percentage difference if available and meaningful, to provide a more quantitative initial statement beyond just the p-value, when discussing the initial finding of higher IRF5 expression in females, For example, 'IRF5 gene and protein levels were significantly elevated in the AT of overweight/obese females compared to males (p < 0.0001), showing an X-fold increase (or Y% higher expression) in females.
  7. Similarly, in lines 175-176, for the immunofluorescent analysis, explicitly state the fold increase in protein levels in the text, as it's a key finding from Figure 1B. For instance, 'Immunofluorescent analysis revealed a significant two-fold increase in IRF5 protein levels in AT of females compared to males (Figure 1B-C).
  8. In table to please change HOMAIR into HOMA-IR. HBA1C should be HbA1c
  9. In table 2, authors should add the measuring units for the metabolic markers.
  10. In lines 218-219, please change the table 1 title as correlation between IRF5 expression in adipose tissue and inflammatory markers in male and female subjects or IRF5 expression–inflammation link in adipose tissue: male vs. female
  11. In lines 224-248, while Table 1 provides detailed correlations, summarizing the most striking sex-specific differences in correlations within the text of sections 3.3 and 3.4 more prominently would guide the reader. For example, explicitly highlight the strongest positive correlations for males (e.g., IL-2, IL-8, TNF-α, IRAK1) and for females (e.g., IL-10, IRF4, TLRs, CD68, CD86, CCL7) immediately after stating the general patterns. This would make the key takeaways from Table 1 more apparent in the narrative.
  12. The authors interpret correlative findings as evidence of regulatory interactions, which may be misleading. For example, a positive correlation between IRF5 and IL-6 does not establish directionality or causality, particularly in a cross-sectional human dataset. Revise the language throughout the manuscript to avoid causal overstatements (e.g., replace “drives” or “regulates” with “is associated with”).
  13. Several sentences in the results section are unnecessarily long and difficult to follow. Shorten and clarify these sentences for improved readability.

Author Response

Reviewer 3

Comments and Suggestions for Authors

This manuscript by Kochumon et al. investigates the sex-specific expression of IRF5 in human adipose tissue and its associations with inflammation and insulin resistance in obese individuals. The authors analyze gene expression data from adipose biopsies, highlighting differential associations between IRF5 and metabolic parameters in females versus males. While the manuscript provides interesting observations, several key concerns related to interpretation, analytical depth, and translational context need to be addressed before the study can be considered for publication.

We would like to thank the reviewer for the compliments and insights. We appreciate the positive feedback.

Major Comments

  1. The introduction lacks a detailed rationale for focusing specifically on IRF5 among the various immune regulators involved in adipose inflammation.

Authors’ response: We thank the reviewer for this insightful suggestion. In the revised introduction, we have expanded the rationale for focusing on IRF5 among the many immune regulators involved in adipose tissue inflammation.

  1. The background does not clearly explain the biological plausibility for sex differences in IRF5 expression or signaling.

Authors’ response: We thank the reviewer for this suggestion.

  1. Expand the introduction to justify IRF5 selection and elaborate on possible biological pathways and their involvements in inflammations.

Authors response: We appreciate the reviewer’s valuable feedback. In the revised introduction, we have expanded the rationale for selecting IRF5 and elaborated on its biological pathways involved in inflammation.

  1. Please provide the Information about the sample collection (e.g., fasting status, comorbidities, or medications) which could confound IRF5 expression. These can strongly affect IRF5 expression and insulin sensitivity.

Authors’ response: We appreciate the reviewer’s insightful comment. In our study, blood and adipose tissue biopsy samples were collected from all participants after an overnight fast of at least 10 hours to minimize metabolic variability. To further reduce potential confounding factors that could influence IRF5 expression and insulin sensitivity, individuals with a history of cardiovascular, pulmonary, renal, hepatic, or hematological diseases, autoimmune disorders, malignancies, pregnancy, type 1 diabetes, or type 2 diabetes were excluded.

  1. Line 72: Clarify the meaning of “metabolic disease” obesity.

      Authors’ response: We corrected the sentences, please see Line 93-94, red font

  1. In lines 173-174, authors should consider adding the specific fold change or percentage difference if available and meaningful, to provide a more quantitative initial statement beyond just the p-value, when discussing the initial finding of higher IRF5 expression in females, For example, 'IRF5 gene and protein levels were significantly elevated in the AT of overweight/obese females compared to males (p < 0.0001), showing an X-fold increase (or Y% higher expression) in females.

Authors’ response: We thank the reviewer for the notification. We corrected the sentence showing the fold change, as suggested, please see lines 100-106.

  1. Similarly, in lines 175-176, for the immunofluorescent analysis, explicitly state the fold increase in protein levels in the text, as it's a key finding from Figure 1B. For instance, 'Immunofluorescent analysis revealed a significant two-fold increase in IRF5 protein levels in AT of females compared to males (Figure 1B-C).

Authors’ response: We thank the reviewer for the clarifications.

  1. In table to please change HOMAIR into HOMA-IR. HBA1C should be HbA1c.

Authors’ response: We thank the reviewer for the notification. Table 2 was corrected.

  1. In table 2, authors should add the measuring units for the metabolic markers.

Authors’ response: We thank the reviewer for the notification. Table 2 was corrected.

  1. In lines 218-219, please change the table 1 title as correlation between IRF5 expression in adipose tissue and inflammatory markers in male and female subjects or IRF5 expression–inflammation link in adipose tissue: male vs. female

Authors’ response: We thank the reviewer for the notification; the new title reflects the table correctly. Table 1 title was corrected as suggested. 

  1. In lines 224-248, while Table 1 provides detailed correlations, summarizing the most striking sex-specific differences in correlations within the text of sections 3.3 and 3.4 more prominently would guide the reader. For example, explicitly highlight the strongest positive correlations for males (e.g., IL-2, IL-8, TNF-α, IRAK1) and for females (e.g., IL-10, IRF4, TLRs, CD68, CD86, CCL7) immediately after stating the general patterns. This would make the key takeaways from Table 1 more apparent in the narrative.

Authors’ response: We thank the reviewer for the notification; however, we find adding this sentence will disturb the flow of the paragraph and each of the factors has to be in full name, besides the appreciated gene name. This will add repetition to the sections.  

  1. The authors interpret correlative findings as evidence of regulatory interactions, which may be misleading. For example, a positive correlation between IRF5 and IL-6 does not establish directionality or causality, particularly in a cross-sectional human dataset. Revise the language throughout the manuscript to avoid causal overstatements (e.g., replace “drives” or “regulates” with “is associated with”).

Authors’ response: We thank the reviewer for this important observation. We agree that correlation does not imply causation, especially in the context of a cross-sectional human dataset. In response, we have carefully revised the manuscript to avoid causal overstatements.

  1. Several sentences in the results section are unnecessarily long and difficult to follow. Shorten and clarify these sentences for improved readability.

Authors’ response: We thank the reviewer for this valuable suggestion. We have carefully revised the Results section to shorten overly long sentences and improve clarity.

Round 2

Reviewer 2 Report

Comments and Suggestions for Authors

The authors have thoroughly revised the manuscript and addressed all concerns

Reviewer 3 Report

Comments and Suggestions for Authors

Dear Editor,

I have re-evaluated the revised version of the manuscript entitled “Sex-Specific Differences in Adipose IRF5 Expression and Its Association with Inflammation and Insulin Resistance in Obesity” submitted to IJMS. I am pleased to note that the authors have adequately and comprehensively addressed all the comments and concerns raised in my previous review.

The revisions have substantially improved the manuscript in terms of clarity, presentation, and scientific rigor. 

Given the quality of the work and the satisfactory response to all reviewer queries, I recommend the manuscript for acceptance in its current form.

Sincerely,